# Succinct Network Channel and Spatial Pruning via Discrete Variable QCQP

## Abstract

Reducing the heavy computational cost of large convolutional neural networks is crucial when deploying the networks to resource-constrained environments. In this context, recent works propose channel pruning via greedy channel selection to achieve practical acceleration and memory footprint reduction. We first show this channel-wise approach ignores the inherent quadratic coupling between channels in the neighboring layers and cannot safely remove inactive weights during the pruning procedure. Furthermore, we show that these pruning methods cannot guarantee the given resource constraints are satisfied and cause discrepancy with the true objective. To this end, we formulate a principled optimization framework with discrete variable QCQP, which provably prevents any inactive weights and enables the exact guarantee of meeting the resource constraints in terms of FLOPs and memory. Also, we extend the pruning granularity beyond channels and jointly prune individual 2D convolution filters spatially for greater efficiency. Our experiments show competitive pruning results under the target resource constraints on CIFAR-10 and ImageNet datasets on various network architectures.

## 1 Introduction

Deep neural networks are the bedrock of artificial intelligence tasks such as object detection, speech recognition, and natural language processing (Redmon & Farhadi, 2018; Chorowski et al., 2015; Devlin et al., 2019). While modern networks have hundreds of millions to billions of parameters to train, it has been recently shown that these parameters are highly redundant and can be pruned without significant loss in accuracy (Han et al., 2015; Guo et al., 2016). This discovery has led practitioners to desire training and running the models on resource-constrained mobile devices, provoking a large body of research on network pruning.

Unstructured pruning, however, does not directly lead to any practical acceleration or memory footprint reduction due to poor data locality (Wen et al., 2016), and this motivated research on structured pruning to achieve practical usage under limited resource budgets. To this end, a line of research on channel pruning considers completely pruning the convolution filters along the input and output channel dimensions, where the resulting pruned model becomes a smaller dense network suited for practical acceleration and memory footprint reduction (Li et al., 2017; Luo et al., 2017; He et al., 2019; Wen et al., 2016; He et al., 2018a).

However, existing channel pruning methods perform the pruning operations with a greedy approach and does not consider the inherent quadratic coupling between channels in the neighboring layers. Although these methods are easy to model and optimize, they cannot safely remove inactive weights during the pruning procedure, suffer from discrepancies with the true objective, and prohibit the strict satisfaction of the required resource constraints during the pruning process.

The ability to specify hard target resource constraints into the pruning optimization process is important since this allows the user to run the pruning and optional finetuning process only once. When the pruning process ignores the target specifications, the users may need to apply multiple rounds of pruning and finetuning until the specifications are eventually met, resulting in an extra computation overhead (Han et al., 2015; He et al., 2018a; Liu et al., 2017).

In this paper, we formulate a principled optimization problem that prunes the network layer channels while respecting the quadratic coupling and exactly satisfying the user-specified FLOPs and

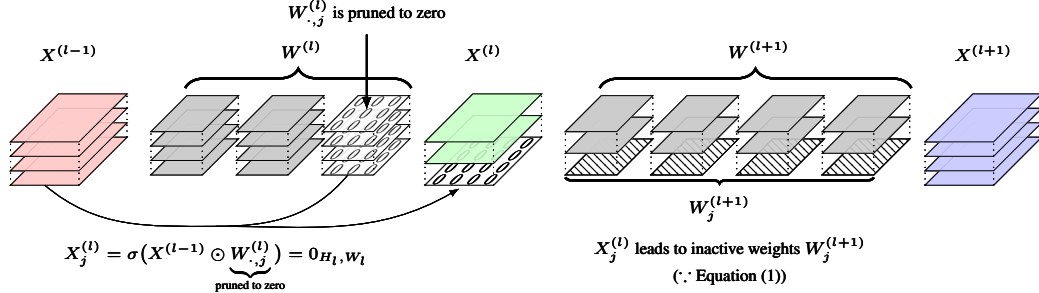

Figure 1: Illustration of a channel pruning procedure that leads to inactive weights. When $j$-th output channel of $l$-th convolution weights $W_{\cdot,j}^{(l)}$ is pruned, *i.e.* $W_{\cdot,j}^{(l)} = 0_{C_{l-1},K_l,K_l}$, then the $j$-th feature map of $l$-th layer $X_j^{(l)}$ should also be 0. Consequently, $X_j^{(l)}$ yields inactive weights $W_j^{(l+1)}$. Note that we use $W_{\cdot,j}^{(l)}$ to denote the tensor $W_{\cdot,j,\cdot,\cdot}^{(l)}$, following the indexing rules of NumPy (Van Der Walt et al., 2011).

memory constraints. This new formulation leads to an interesting discrete variable QCQP (Quadratic Constrained Quadratic Program) optimization problem, which directly maximizes the importance of neurons in the pruned network under the specified resource constraints. Also, we increase the pruning granularity beyond channels and jointly prune individual 2D convolution filters spatially for greater efficiency. Furthermore, we generalize our formulation to cover nonsequential convolution operations, such as skip connections, and propose a principled optimization framework for handling various architectural implementations of skip connections in ResNet (He et al., 2016). Our experiments on CIFAR-10 and ImageNet datasets show the state of the art results compared to other channel pruning methods that start from pretrained networks.

## 2 MOTIVATION

In this section, we first discuss the motivation of our method concretely. Suppose the weights in a sequential CNN form a sequence of 4-D tensors, $W^{(l)} \in \mathbb{R}^{C_{l-1} \times C_l \times K_l \times K_l} \ \forall l \in [L]$ where $C_{l-1}, C_l$, and $K_l$ represent the number of input channels, the number of output channels, and the filter size of $l$-th convolution weight tensor, respectively. We denote the feature map after $l$-th convolution as $X^{(l)} \in \mathbb{R}^{C_l \times H_l \times W_l}$. Concretely, $X_j^{(l)} = \sigma(X^{(l-1)} \odot W_{\cdot,j}^{(l)}) = \sigma(\sum_{i=1}^{C_{l-1}} X_i^{(l-1)} * W_{i,j}^{(l)})$, where $\sigma$ is the activation function, $*$ denotes 2-D convolution operation, and $\odot$ denotes the sum of channel-wise 2-D convolutions. Now consider pruning these weights in channel-wise direction. We show that naive channel-wise pruning methods prevent exact specification of the target resource constraints due to unpruned inactive weights and deviate away from the true objective by ignoring quadratic coupling between channels in the neighboring layers.

### 2.1 INACTIVE WEIGHTS

According to Han et al. (2015), network pruning produces dead neurons with zero input or output connections. These dead neurons cause *inactive* weights[1], which do not affect the final output activations of the pruned network. These inactive weights may not be excluded automatically through the standard pruning procedure and require additional post-processing which relies on ad-hoc heuristics. For example, Figure 1 shows a standard channel pruning procedure that deletes weights across the output channel direction but fails to prune the inactive weights. Concretely, deletion of weights on $j$-th output channel of $l$-th convolution layer leads to $W_{\cdot,j}^{(l)} = 0_{C_{l-1},K_l,K_l}$. Then, $X_j^{(l)}$ becomes a dead neuron since $X_j^{(l)} = \sigma(X^{(l-1)} \odot W_{\cdot,j}^{(l)}) = \sigma(\sum_{i=1}^{C_{l-1}} X_i^{(l-1)} * W_{i,j}^{(l)}) = 0_{C_l,H_l,W_l}$.

---

[1]Rigorous mathematical definition of inactive weights is provided in Supplementary material D.

The convolution operation on the dead neuron results in a trivially zero output, as below:

$$X_p^{(l+1)} = \sigma\left(\sum_{i=1}^{C_l} X_i^{(l)} * W_{i,p}^{(l+1)}\right) = \sigma\left(\sum_{i=1}^{C_l} \mathbb{1}_{i \neq j} X_i^{(l)} * W_{i,p}^{(l+1)} + \underbrace{\underbrace{X_j^{(l)}}_{\text{dead}} * \underbrace{W_{j,p}^{(l+1)}}_{\text{inactive}}}_{=0_{H_{l+1}, W_{l+1}}}\right). \quad (1)$$

Equation (1) shows that the dead neuron $X_j^{(l)}$ causes weights $W_{j,p}^{(l+1)}, \forall p \in [C_{l+1}]$ to be inactive. Such inactive weights do not account for the actual resource usage, even when they remain in the pruned network, which prevents the exact modeling of the user-specified hard resource constraints (FLOPs and network size). Furthermore, inactive weights unpruned during the pruning procedure are a bigger problem for nonsequential convolutional networks due to their skip connections. To address this problem, we introduce a quadratic optimization-based algorithm that provably eliminates all the inactive weights during the pruning procedure.

## 2.2 QUADRATIC COUPLING

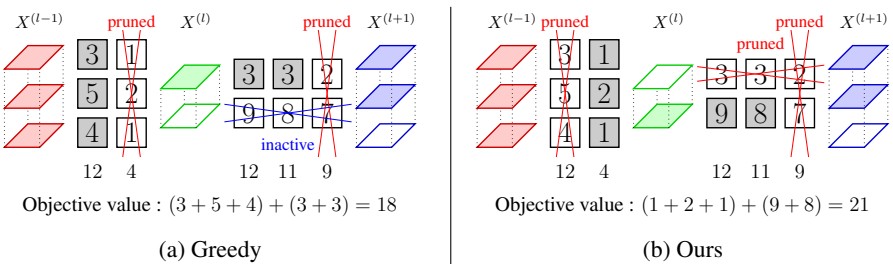

(a) Greedy

(b) Ours

Figure 2: A comparison of the greedy channel pruning method and our pruning method. Parallelograms represent feature maps and squares represent 2-D filters of convolution weights. Gray squares are filters which account for the objective. The numbers on each squares represent the absolute sum of weights in the filter.

Existing channel pruning methods remove channels according to their importance. However, measuring a channel's contribution to the network should also take into account the channels in the neighboring layers, as illustrated in Figure 2. In the example, we define the importance of a channel as the absolute sum of weights in the channel, as in Li et al. (2017), and assume the objective is to maximize the absolute sum of weights in the whole pruned network, excluding the inactive weights. We compare two different channel pruning methods: (a) a standard channel pruning method that greedily prunes each channel independently, and (b) our pruning method that considers the effect of the channels in neighboring layers when pruning. As a result of running each pruning algorithms, (a) will prune the second output channel of the first convolution and the third output channel of the second convolution, and (b) will prune the first output channel of the first convolution, the third output channel of the second convolution, and the first input channel of the second convolution. The objective values for each pruned networks are (a) 18 and (b) 21, respectively.

This shows that the coupling effect of the channels in neighboring layers directly affects the objective values, and finally results in a performance gap between (a) and (b). We call this coupling relationship as the *quadratic coupling* between the neighboring layers and formulate the contributions to the objective by quadratic terms of neighboring channel activations. To address this quadratic coupling, we propose a channel pruning method based on the QCQP framework with importance evaluation respecting both the input and the output channels.

## 3 METHOD

In this section, we first propose our discrete QCQP formulation of channel pruning for the sequential convolutional neural networks (CNNs). Then, we present an extended version of our formulation for joint channel and shape pruning of 2D convolution filters. The generalization to the nonsequential convolution (skip addition and skip concatenation) is introduced in Supplementary material A.

### 3.1 FORMULATION OF CHANNEL PRUNING FOR SEQUENTIAL CNNS

To capture the importance of weights in $W^{(l)}$, we define the *importance tensor* as $I^{(l)} \in \mathbb{R}_+^{C_{l-1} \times C_l \times K_l \times K_l}$. Following the protocol of Han et al. (2015); Guo et al. (2016), we set $I^{(l)} = \gamma_l |W^{(l)}|$ where $\gamma_l$ is the $\ell_2$ normalizing factor in $l$-th layer or $\|\text{vec}(W^{(l)})\|^{-1}$. Then, we define the binary pruning mask as $A^{(l)} \in \{0,1\}^{C_{l-1} \times C_l \times K_l \times K_l}$. For channel pruning in sequential CNNs, we define *channel activation* $r^{(l)} \in \{0,1\}^{C_l}$ to indicate which indices of channels remain in the $l$-th layer of the pruned network. Then, the weights in $W_{i,j}^{(l)}$ are active if and only if $r_i^{(l-1)} r_j^{(l)} = 1$, which leads to $A_{i,j}^{(l)} = r_i^{(l-1)} r_j^{(l)} J_{K_l}$. For example, in Figure 2b, $r^{(l-1)} = [1,1,1]^\mathsf{T}$, $r^{(l)} = [0,1]^\mathsf{T}$, and $r^{(l+1)} = [1,1,0]^\mathsf{T}$, therefore, $A^{(l)} = \begin{bmatrix} 0 & 1 \\ 0 & 1 \\ 0 & 1 \end{bmatrix} \otimes J_{K_l}$ and $A^{(l+1)} = \begin{bmatrix} 0 & 0 & 0 \\ 1 & 1 & 0 \end{bmatrix} \otimes J_{K_{l+1}}$.

We wish to directly maximize the sum of the importance of active weights after the pruning procedure under given resource constraints : 1) FLOPs, 2) memory, and 3) network size. Concretely, our optimization problem is [2]

$$\underset{r^{(0:L)}}{\text{maximize}} \quad \sum_{l=1}^{L} \left\langle I^{(l)}, A^{(l)} \right\rangle \tag{2}$$

$$\text{subject to} \quad \sum_{l=0}^{L} a_l \left\| r^{(l)} \right\|_1 + \sum_{l=1}^{L} b_l \left\| A^{(l)} \right\|_1 \leq M$$

$$A^{(l)} = r^{(l-1)} r^{(l)\mathsf{T}} \otimes J_{K_l} \quad \forall l \in [L]$$

$$r^{(l)} \in \{0,1\}^{C_l}.$$

In our formulation, the actual resource usage of the pruned network is exactly computed by specifying the number of channels in the pruned network ($= \|r^{(l)}\|_1$) and the pruning mask sparsity ($= \|A^{(l)}\|_1$) in each layer. Concretely, the left hand side of the inequality in the first constraint in Equation (2) indicates the actual resource usage. Table 1 shows $a_l, b_l$ terms used for computing usage of each resource. Note that this optimization problem is a discrete nonconvex QCQP of the channel activations $[r^{(0)}, \ldots, r^{(L)}]$, where the objective, which is the same with the objective in Section 2.2, respects the quadratic coupling of channel activations ($= r^{(l)}$). Please refer to Supplementary material E for the details on the standard QCQP form of Equation (2).

### 3.2 FORMULATION OF JOINT CHANNEL AND SPATIAL PRUNING

For further efficiency, we increase the pruning granularity to additionally perform spatial pruning in 2-D convolution filters. Concretely, we prune by each weight vector across the input channel direction instead of each channel to perform channel and spatial pruning processes simultaneously.

| Resource constraint (M) | $a_l$ | $b_l$ |
|---|---|---|
| Network size | 0 | 1 |
| Memory | $H_l W_l$ | 1 |
| FLOPs | 0 | $H_l W_l$ |

Table 1: Resource constraints and the corresponding $a_l, b_l$ values.

First, we define the *shape column* $W_{\cdot,j,a,b}^{(l)}$ by the vector of weights at spatial position $(a,b)$ of a 2-D convolution filter along the $j$-th output channel dimension. Then, we define *shape column activation* $q^{(l)} \in \{0,1\}^{C_l \times K_l \times K_l}$ to indicate which shape columns in the $l$-th convolution layer remain in the pruned network. Figure 3 shows the illustration of each variables.

Note that this definition induces constraints on the channel activation variables. In detail, the $j$-th output channel activation in $l$-th layer is set if and only if at least one shape column activation in the $j$-th output channel is set. Concretely, the new formulation should include the constraints $r_j^{(l)} \leq \sum_{a,b} q_{j,a,b}^{(l)}$ and $q_{j,a,b}^{(l)} \leq r_j^{(l)} \; \forall a, b$.

---

[2] $\otimes$ denotes the outer product of tensors and $J_n$ is a $n$-by-$n$ matrix of ones.

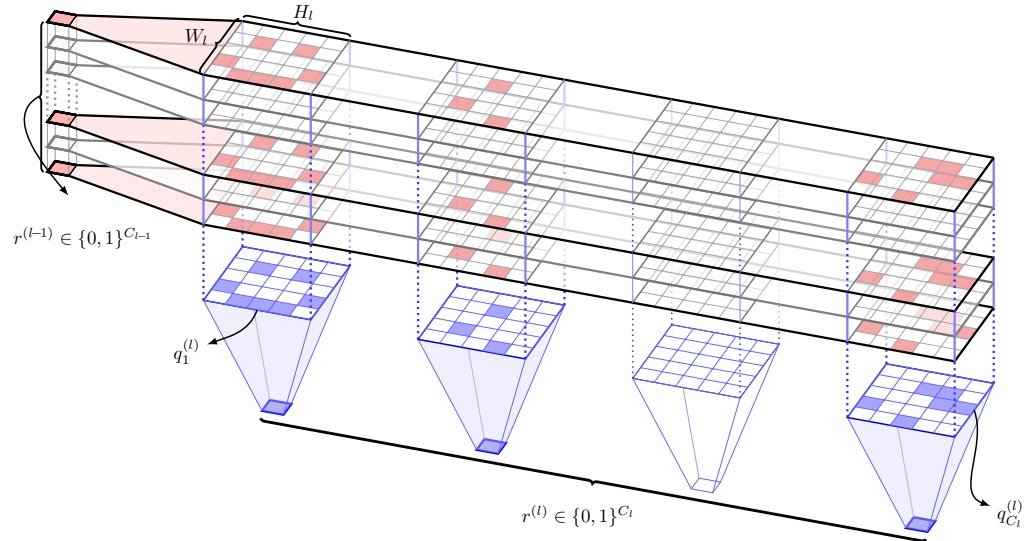

Figure 3: Input channel activation $\left(= r^{(l-1)}\right)$, shape column activation $\left(= q^{(l)}\right)$, and the corresponding mask $\left(= A^{(l)}\right)$ for $l$-th convolution layer, where $A^{(l)} = r^{(l-1)} \otimes q^{(l)}$.

We now reformulate the optimization problem to include the shape column activation variables. Again, we aim to maximize the sum of the importance of active weights after pruning under the given resource constraints. Then, our optimization problem for simultaneous channel and spatial pruning is

$$\underset{r^{(0:L)}, q^{(1:L)}}{\text{maximize}} \quad \sum_{l=1}^{L} \left\langle I^{(l)}, A^{(l)} \right\rangle \tag{3}$$

$$\text{subject to} \quad \sum_{l=0}^{L} a_l \left\| r^{(l)} \right\|_1 + \sum_{l=1}^{L} b_l \left\| A^{(l)} \right\|_1 \leq M$$

$$r_j^{(l)} \leq \sum_{a,b} q_{j,a,b}^{(l)} \quad \text{and} \quad q_{j,a,b}^{(l)} \leq r_j^{(l)} \quad \forall l, j, a, b$$

$$A^{(l)} = r^{(l-1)} \otimes q^{(l)} \quad \forall l$$

$$r^{(l)} \in \{0, 1\}^{C_l} \text{ and } q^{(l)} \in \{0, 1\}^{C_l \times K_l \times K_l} \quad \forall l \in [L].$$

We note that this optimization problem is also a discrete nonconvex QCQP. The details on the standard QCQP form of Equation (3) is provided in Supplementary material E. Furthermore, Proposition 1 below shows that the constraints in Equation (2) and Equation (3) provably eliminate any unpruned inactive weights and accurately model the resource usage as well as the objective of the pruned network. Also, Proposition 1 can be generalized to nonsequential networks with skip addition. The generalization and the proofs are given in Supplementary material D.

**Proposition 1.** *Optimizing over the input and output channel activation variables $r^{(0:L)}$ and shape column activation variables $q^{(1:L)}$ under the constraints in Equation (3) provably removes any inactive weights in the pruned network, guaranteeing exact computation of 1) resource usage and 2) the sum of the importance of active weights in the pruned network.*

### 3.3 OPTIMIZATION

Concretely, Equation (2) and Equation (3) fall into the category of binary Mixed Integer Quadratic Constraint Quadratic Programming (MIQCQP). We solve these discrete QCQP problems with the CPLEX library (INC, 1993), which provides MIQCQP solvers based on the branch and cut technique. However, the branch and cut algorithm can lead to exponential search time (Mitchell, 2002) on large problems. Therefore, we provide a practical alternative utilizing a block coordinate descent style optimization method in Supplementary material B.

## 4 RELATED WORKS

**Importance of channels**   Most of the channel pruning methods prune away the least important channels with a simple greedy approach, and the evaluation method for the importance of channels has been the main research problem (Molchanov et al., 2017; 2019; Liu et al., 2019). Channel pruning is divided into two major branches according to the method of evaluating the importance of channels: the *trainable-importance* method, which evaluates the importance of channels while training the whole network from scratch, and the *fixed-importance* method, which directly evaluates the importance of channels on the pretrained network. Trainable-importance channel pruning methods include the regularizer-based methods with group sparsity regularizers (Wen et al., 2016; Alvarez & Salzmann, 2016; Yang et al., 2019; Liu et al., 2017; Louizos et al., 2018; Liu et al., 2017; Gordon et al., 2018) and data-driven channel pruning methods (Kang & Han, 2020; You et al., 2019). Fixed-importance channel pruning methods first prune away most of the weights and then finetune the significantly smaller pruned network (Molchanov et al., 2017; 2019; Hu et al., 2016; He et al., 2018a; Li et al., 2017; He et al., 2019; Luo et al., 2017). As a result, fixed-importance methods are much more efficient than the trainable-importance channel pruning methods in terms of computational cost and memory as trainable-importance methods have to train the whole unpruned network. Our framework is on the line of fixed-importance channel pruning works.

**Predefined target structure and Automatic target structure**   Layer-wise channel pruning methods (Li et al., 2017; He et al., 2019), which perform pruning operations per each layer independently, require users to predefine the target pruned structure. Also, LCCL(Dong et al., 2017) exploits a predefined low-cost network to improve inference time. Another line of research finds the appropriate target structure automatically (He et al., 2018b; Yang et al., 2018; Liu et al., 2019; Molchanov et al., 2017). Our method also finds the target structure automatically under the explicit target resource constraints.

**Dynamic pruning and static pruning**   Dynamic pruning has a different network structure depending on the input during inference time, while static pruning has a fixed network structure during inference time. CGNet (Hua et al., 2019) dynamically identifies unnecessary features to reduce the computation, and FBS (Gao et al., 2019) dynamically skip computations on the unimportant channels. Our framework is static pruning and has a fixed network structure during inference time.

**Quadratic coupling**   CCP (Peng et al., 2019) formulates a QP (quadratic formulation) to consider the quadratic coupling between channels in the same layer under layer-wise constraints on the maximum number of channels. On the other hand, our formulation considers the quadratic coupling between channels in the neighboring layers under the target resource constraints.

**Channel pruning in nonsequential blocks**   Many network architectures contain nonsequential convolution operations, such as skip addition (He et al., 2016; Sandler et al., 2018; Tan & Le, 2019) and skip concatenation (Huang et al., 2017). Since these network architectures outperform sequential network architectures, pruning a network with nonsequential convolution operations is crucial. However, most channel pruning methods (Liu et al., 2017; He et al., 2018a; 2019; Molchanov et al., 2017; 2019) do not consider the nonsequential convolution operations and use the same method from the sequential network architecture. However, channel pruning methods ignorant of nonsequential convolution operations may result in the misalignment of feature maps connected by skip connections (You et al., 2019). GBN (You et al., 2019) forces parameters connected by a nonsequential convolution operation to share the same pruning pattern to solve this misalignment problem. In contrast, our formulation does not require strict pattern sharing. This flexibility allows for our methods to delete more channels under given constraints.

**Spatial pruning of convolution filters**   Spatial pruning methods aim to prune convolution filters along the channel dimension for inference efficiency. Spatial pruning methods manually define the spatial patterns of filters (Lebedev & Lempitsky, 2016; Anwar et al., 2017) or optimize spatial patterns of filters with group sparse regularizers (Wen et al., 2016; Lebedev & Lempitsky, 2016). Among these works, Lebedev & Lempitsky (2016) empirically demonstrates that enforcing sparse spatial patterns in 2-D filters along the input channel leads to great speed-up during inference time

using group sparse convolution operations (Chellapilla et al., 2006). Our proposed method enforces the spatial patterns in 2-D filters as in Lebedev & Lempitsky (2016) for speed-up in inference.

## 5 EXPERIMENTS

We compare the classification accuracy of the pruned network against several pruning baselines on CIFAR-10 (Krizhevsky et al., 2009) and ImageNet (Russakovsky et al., 2015) datasets using various ResNet architectures (He et al., 2016), DenseNet-40 (Huang et al., 2017), and VGG-16 (Simonyan & Zisserman, 2015). Note that most pruning baselines apply an iterative pruning procedure, which repeatedly alternates between network pruning and finetuning until the target resource constraints are satisfied (Han et al., 2015; He et al., 2018a; Liu et al., 2017; Yang et al., 2018). In contrast, since our methods explicitly include the target resource constraint to the optimization framework, we only need one round of pruning and finetuning.

### 5.1 EXPERIMENTAL SETTINGS

We follow the 'smaller-norm-less-important' criterion (Ye et al., 2018; Liu et al., 2017), which evaluates the importance of weights as the absolute value of weight (Han et al., 2015; Guo et al., 2016). We assume the network size and FLOPs reduction are linearly proportional to the sparsity in shape column activations, as empirically shown in Lebedev & Lempitsky (2016). Also, we ignore the extra memory overhead for storing the shape column activations due to its negligible size compared to the total network size. In the experiment tables, FLOPs and the network size of the pruned network are computed according to the resource specifications in Equations (2) and (3). Also, 'Pruning ratio' in the tables denotes the ratio of pruned weights among the total weights in baseline networks. 'ours-c' and 'ours-cs' in the tables denote our method with channel pruning and our method with both the channel and spatial pruning, respectively.

| Network | Method | Baseline acc | FLOPs | | | Network size | | |
|---|---|---|---|---|---|---|---|---|
| | | | Pruned acc↑ | Acc drop↓ | FLOPs(%)↓ | Pruned acc↑ | Acc drop↓ | Pruning ratio(%)↑ |
| ResNet-20 | FPGM (He et al., 2019) | 92.21 (0.18) | 91.26 (0.24) | 0.95 | **46.0** | 91.26 (0.24) | 0.95 | 54.0 |
| | ours-c | 92.21 (0.18) | 90.96 (0.15) | 1.25 | **46.0** | 91.26 (0.18) | 0.95 | **54.1** |
| | ours-cs | 92.21 (0.18) | **91.70 (0.18)** | **0.51** | **46.0** | **92.02 (0.10)** | **0.19** | 54.0 |
| | LCCL (Dong et al., 2017) | 92.74 | 91.68 | 1.06 | 62.1 | 91.68 | 1.06 | 33.1 |
| | SFP (He et al., 2018a) | 92.20 (0.18) | 90.83 (0.31) | 1.37 | **57.8** | 90.83 (0.31) | 1.37 | 42.2 |
| | FPGM (He et al., 2019) | 92.21 (0.18) | 91.72 (0.20) | 0.49 | **57.8** | 91.72 (0.20) | 0.49 | 42.2 |
| | ours-c | 92.21 (0.18) | 91.74 (0.20) | 0.47 | 58.3 | 92.27 (0.17). | -0.06 | **42.3** |
| | ours-cs | 92.21 (0.18) | **92.26 (0.10)** | **-0.05** | 57.8 | **92.35 (0.10)** | **-0.14** | 42.2 |
| ResNet-32 | FPGM (He et al., 2019) | 92.88 (0.86) | 91.96 (0.76) | 0.92 | **46.8** | 91.96 (0.76) | 0.92 | **53.2** |
| | ours-c | 92.88 (0.86) | 91.98 (0.42) | 0.90 | **46.8** | 92.22 (1.02) | 0.66 | **53.2** |
| | ours-cs | 92.88 (0.86) | **92.33 (0.41)** | **0.55** | 47.0 | **92.78 (0.97)** | **0.10** | **53.2** |
| | LCCL (Dong et al., 2017) | 92.33 | 90.74 | 1.59 | 69.0 | 90.74 | 1.59 | 37.5 |
| | SFP (He et al., 2018a) | 92.63 (0.70) | 92.08 (0.08) | 0.55 | 58.5 | 92.08 (0.08) | 0.55 | 41.5 |
| | FPGM (He et al., 2019) | 92.88 (0.86) | 92.51 (0.90) | 0.37 | 58.5 | 92.51 (0.90) | 0.37 | 41.5 |
| | ours-c | 92.88 (0.86) | 92.52 (0.46) | 0.36 | **57.2** | 92.42 (0.77) | 0.46 | **42.7** |
| | ours-cs | 92.88 (0.86) | **92.80 (0.61)** | **0.08** | 57.9 | **92.83 (0.83)** | **0.05** | **42.7** |
| ResNet-56 | SFP (He et al., 2018a) | 93.59 (0.58) | 92.26 (0.31) | 1.33 | 47.5 | 92.26 (0.31) | 1.33 | 52.6 |
| | FPGM (He et al., 2019) | 93.59 (0.58) | 93.49 (0.13) | 0.10 | 47.5 | 93.49 (0.13) | 0.10 | 52.6 |
| | CCP (Peng et al., 2019) | 93.50 | 93.42 | 0.08 | **47.4** | - | - | - |
| | AMC (He et al., 2018b) | 92.8 | 91.9 | 0.9 | 50.0 | - | - | - |
| | SCP (Kang & Han, 2020) | 93.69 | 93.23 | 0.46 | 48.5 | 93.23 | 0.46 | 51.5 |
| | ours-c | 93.59 (0.58) | 93.36 (0.68) | 0.23 | **47.4** | 93.37 (0.96) | 0.22 | **52.7** |
| | ours-cs | 93.59 (0.58) | **93.59 (0.36)** | **0.00** | **47.4** | **93.69 (0.69)** | **-0.10** | 52.6 |
| DenseNet-40 | SCP (Kang & Han, 2020) | 94.39 | 93.77 | **0.62** | **29.2** | - | - | - |
| | ours-c | 95.01 | 93.80 | 1.21 | **29.2** | - | - | - |
| | ours-cs | 95.01 | **94.25** | 0.76 | **29.2** | - | - | - |
| | slimming (Liu et al., 2017) | 93.89 | 94.35 | **-0.46** | **45.0** | - | - | - |
| | ours-c | 95.01 | 94.38 | 0.63 | **45.0** | - | - | - |
| | ours-cs | 95.01 | **94.85** | 0.16 | **45.0** | - | - | - |
| | slimming (Liu et al., 2017) | 93.89 | 94.81 | **-0.92** | 71.6 | - | - | - |
| | ours-c | 95.01 | 94.82 | 0.19 | **71.0** | - | - | - |
| | ours-cs | 95.01 | **95.02** | -0.01 | **71.0** | - | - | - |

Table 2: Pruned accuracy and accuracy drop from the baseline network at given FLOPs (left) and pruning ratios (right) on various network architectures (ResNet-20,32,56 and DenseNet-40) at CIFAR-10.

## 5.2 CIFAR-10

CIFAR-10 dataset has 10 different classes with $5k$ training images and $1k$ test images per each class Krizhevsky et al. (2009). In CIFAR-10 experiments, we evaluate our methods on various network architectures: ResNet-20, 32, 56, and DenseNet-40. We provide implementation of the details for the experiments in Supplementary material C. We show the experiment results of pruning under FLOPs constraints in the left column of Table 2 and under final network size constraints in the right column of Table 2.

On the FLOPs experiments in the left column of Table 2, 'ours-c' shows comparable results against FPGM, which is the previous state of the art method, on ResNet-20, 32, and 56. Moreover, 'ours-cs' significantly outperforms both 'ours-c' and FPGM on the same architectures showing the state of the art performance. Also, 'ours-c' shows comparable results against slimming (Liu et al., 2017) and SCP (Kang & Han, 2020), while 'ours-cs' outperforms existing baselines by a large margin on DenseNet-40.

On the network size experiments in the right column of Table 2, 'ours-c' shows results competitive to FPGM and SCP, while 'ours-cs' again achieves the state of the art performance on ResNet-20, 32, and 56. Notably, in ResNet-56, 'ours-cs' achieves a minimal accuracy drop of $-0.10$ with the pruning ratio of $52.6\%$. These results show simultaneous channel and spatial pruning produces more efficient networks with better performance compared to other channel pruning methods on CIFAR-10.

## 5.3 IMAGENET

ILSVRC-2012 (Russakovsky et al., 2015) is a large-scale dataset with $1000$ classes that comes with $1.28M$ training images and $50k$ validation images. We conduct our methods under the fixed FLOPs constraint on ResNet-18,50, and VGG-16. For more implementation details of the ImageNet experiments, refer to Supplementary material C. Table 3 shows the experiment results on ImageNet. In ResNet-50, 'ours-c' and 'ours-cs' achieve results comparable to GBN, a trainable-importance channel pruning method which is the previous state of the art, even though our method is a fixed-importance channel pruning method. In particular, top1 pruned accuracy in 'ours-cs' exceeds SFP by $1.32\%$ using a similar number of FLOPs. Both 'ours-cs' and 'ours-c' clearly outperform FPGM in ResNet-50. Also, 'ours-c' and 'ours-cs' show significantly better performance than Molchanov et al. (2017) on VGG-16.

| Network | Method | Top1 Pruned Acc↑ | Top1 Acc drop↓ | Top5 Pruned Acc↑ | Top5 Acc drop↓ | FLOPs(%)↓ |
|---|---|---|---|---|---|---|
| ResNet-18 | SFP (He et al., 2018a) | 67.10 | 3.18 | 87.78 | 1.85 | **58.2** |
| | FPGM (He et al., 2019) | 68.41 | 1.87 | 88.48 | 1.15 | **58.2** |
| | ours-c | 67.48 | 2.28 | 87.78 | 1.30 | 60.9 |
| | ours-cs | **69.59** | **0.17** | **88.94** | **0.14** | **58.2** |
| | LCCL (Dong et al., 2017) | 66.33 | 3.65 | 86.94 | 2.29 | **65.3** |
| | ours-c | 68.65 | 1.11 | 88.69 | 0.39 | 65.9 |
| | ours-cs | **70.05** | **-0.29** | **89.24** | **-0.16** | 65.9 |
| ResNet-50 | SFP (He et al., 2018a) | 74.61 | 1.54 | 92.06 | 0.81 | 58.3 |
| | FPGM (He et al., 2019) | 75.50 | 0.65 | 92.63 | 0.21 | **57.8** |
| | ours-c | 75.78 | 0.37 | 91.86 | 1.01 | **57.8** |
| | ours-cs | **75.93** | **0.22** | **92.68** | **0.19** | **57.8** |
| | GBN (You et al., 2019) | **76.19** | **-0.31** | 92.83 | **-0.16** | 59.5 |
| | ours-c | 75.89 | 0.26 | **92.84** | 0.03 | 61.5 |
| | ours-cs | 76.00 | 0.15 | 92.76 | 0.11 | **59.0** |
| VGG-16 | Molchanov et al. (2017) | - | - | 84.5 | 5.9 | **51.7** |
| | ours-c | 65.92 | 5.67 | 87.20 | 3.18 | **51.7** |
| | ours-cs | **66.36** | **5.23** | **87.36** | **3.02** | **51.7** |

Table 3: Top1,5 pruned accuracy and accuracy drop from the baseline network at given FLOPs on various network architectures (ResNet-18,50, and VGG-16) at ImageNet.

## 6 CONCLUSION

We present a discrete QCQP based optimization framework for jointly pruning channel and spatial filters under various architecture realizations. Since our methods model the inherent quadratic coupling between channels in the neighboring layers to eliminate any inactive weights during the

pruning procedure, they allow exact modeling of the user-specified resource constraints and enable the direct optimization of the true objective on the pruned network. The experiments show our proposed method significantly outperforms other fixed-importance channel pruning methods, finding smaller and faster networks with the least drop in accuracy.

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
