# OpenReview forum: "Succinct Network Channel and Spatial Pruning via Discrete Variable QCQP"
_ICLR.cc/2021/Conference — Reject_

### Official Review · AnonReviewer1 · 2020-10-21
**Good submission focusing on a valuable topic**

**Rating:** 7
**Confidence:** 4

**Review:**

The authors proposed a pruning method that aims to reduce the parameters and heavy computational cost of large convolutional neural networks (CNNs). According to the comparison experiments performed on several widely-used network fashions, the proposed strategy could help to efficiently reduce the numbers of parameters while maintaining less performance decreasing. I think this study is valuable in both theory and applications.
However, several issues may be further emphasized to make the submission improved:

(1) Were the CNN models deployed in the real resource-constrained environment such as automatic drive hardware with less computational capability? Or could this proposed strategy be applied to reduce the parameters of the CNN models to the level of MobileNet?

(2) Authors determined that the proposed method is useful to tackle several classification tasks, did this method also perform well on the CNN models aimed at segmentation, detection et al. Performance of this kind of models may decrease more than the classification.

---

> ### Author Response · Authors · 2020-11-20
> **Response to Reviewer1**
>
> We thank reviewer 1 for the encouraging comments ("Good submission focusing on a valuable topic", "valuable both in theory and applications") and constructive feedback.
>
> **Q1** : Could this proposed strategy be applied to reduce the parameters of the CNN models to the level of MobileNetV2?
>
> **A** : Thank you for the suggestion. We included our MobileNetV2 experiment results in the table below (see supplementary material G).  '(tuned)' indicates that the normalizing factor ($\gamma_{l}$ in the main paper) is tuned with grid search. A fixed value is used otherwise. Our method shows performance competitive to other pruning methods, [1], [2] and MetaPruning. However, we note that our method is much more efficient than those methods since [1] requires repetitive finetuning steps on the proposed networks, [2] also requires iterative trial and error steps to train the RL agent, and MetaPruning trains PruningNet, which is at least $30$ times bigger than the original model.
>
> |**Network**|**Method**|**Top 1 Pruned Acc$\uparrow$**|**Top1 Acc drop$\uparrow$**|**FLOPs (%)$\downarrow$**|
> |:---:|:--:|:--:|:--:|:--:|
> | MobileNetV2  |       [1]               |            70.9           |          0.9          |       70  |
> |                       |       [2]                |            70.8           |          1.0          |       70   |
> |                       |   MetaPruning   |            **71.2**           |           **0.6**          |       69 |
> ||_________________________________|_________________________________|_________________________________|_________________________________|
> |                       |       ours-c         |            70.8 	 |          1.0          |      **67** |
> |	             |      ours-cs         |           70.2           |          1.6          |       **67** |
> |                       |  ours-c (tuned)   |            71.0           |          0.8         |       **67** |
> |                       | ours-cs (tuned)  |            70.9           |           0.9        |       **67** |
>
>
> **Table** Top1 pruned accuracy and accuracy drop from the baseline network at given FLOPs on MobileNetV2 architecture at ImageNet.
>
>
>
> **Q2** : Authors determined that the proposed method is useful to tackle several classification tasks, did this method also perform well on the CNN models aimed at segmentation, detection et al. Performance of this kind of models may decrease more than the classification.
>
> **A**: Thank you for the suggestion. We applied our pruning method to FCN-32s for segmentation tasks on the PASCAL VOC 2011 dataset. We evaluated the segmentation performance with a widely-used measure, mean Intersection over Union (mIoU), and pruned an original network which has $62.58$ mIoU (see supplementary material I for more details). Our experiment results are shown in the table below. Our method reduces the FLOPs by $27$ \%, with $0.15$ \% mIoU drop on ‘ours-c' and $0.09$ \% mIoU drop on ‘ours-cs'. We will release the pruned FCN-32s model as well as the source code.
>
> | **Network**   |   **FLOPs (%)** $\downarrow$   |     |   **mIoU (%)** $\uparrow$   |  |
> |:--:|:--:|:--:|:--:|:--:|
> |    |   |  **ours-c**   |   **ours-cs**   |   **Original**   |
> |    FCN-32s    |       20         |            49.89            |          49.88         |           -          |
> |	            |      30          |             54.20           |          54.68         |           -          |
> |                       |      40          |            56.10            |          57.24         |           -          |
> |                       |      50          |            58.83            |          58.95         |           -          |
> |                       |      60          |            59.54            |          60.55         |           -          |
> |                      |       70         |            61.65            |         61.88          |           -          |
> |                      |       73         |            62.43            |          62.49         |           -          |
> |                      |      100        |                 -               |             -              |        62.58     |
>
>
> **Table** mIoU (%) of 'ours-c' and 'ours-cs' at different FLOPs on FCN-32s at PascalVOC2016.
>
> **Reference**
>
> [1] NetAdapt: Platform-Aware Neural Network Adaptation for Mobile Applications, ECCV18.
>
> [2] AMC: automl for model compression, ECCV18.

---

### Official Review · AnonReviewer3 · 2020-10-25
**Good motivation and more metrics should be considered**

**Rating:** 5
**Confidence:** 2

**Review:**

Summary：
In this manuscript, a new pruning method is proposed by considering the inherent quadratic constraint between consecutive layers. Without this constraint, inactive weights cannot be safely removed. Even with the same objective function, the optimized result is different, as shown in the motivation section. Based on this observation, the pruning task is models as a QCQP optimization problem. And a faster algorithm to solve this problem is proposed. Moreover, the pruning on filter size can also be modeled as the QCQP problem, making the pruning on both channel and filter size feasible.

Strengths:
-	The paper is well-written and well-motivated. The motivation is reasonable and the proposed method does alleviate the overlooked issue.
-	The results on the CIFAR10 and ImageNet surpass some previous methods.
-	The proposed method does not need iterative pruning procedure as other methods, making it simple to use.
-	The proposed motivation may inspire following works in this area.

Weaknesses:
-	I am not an expert in the area of pruning. I think this motivation is quite good but the results seem to be less impressive. Moreover, I believe the results should be evaluated from more aspects, e.g., the actual latency on target device, the memory consumption during the inference time and the actual network size.
-	The performance is only compared with few methods. And the proposed is not consistently better than other methods. For those inferior results, some analysis should be provided since the results violate the motivation.

I am willing to change my rating according to the feedback from authors and the comments from other reviewers.

---

> ### Author Response · Authors · 2020-11-20
> **Response to Reviewer3**
>
> We thank the reviewer3 for encouraging comments ("well-written and well-motivated", "simple to use", "may inspire following works in this area") and constructive feedback.
>
> **Q1** : Results should be evaluated from more aspects
>
> **A** :  First, we evaluated the pruned ResNet-20 model's actual latency on a machine with 1 GPU (TITAN-XP) using PyTorch. Our pruned model lowers the FLOPs to 46% compared to the original ResNet-20 model. On batch size 512, the inference time of our pruned model and the original model is '18.26ms' and '27.57ms', respectively. This shows our pruned model is 1.51 times faster than the original ResNet-20 model in our environment.
>
> Next, we measure the actual network size during the inference. Our pruned model's network size is 46% compared to the original ResNet-20 model. The network size of the pruned model and the original model is '0.12 MB' and '0.27 MB', respectively.
>
> To sum up, we find that meeting the target FLOPs constraint has increased the inference speed of the pruned network, and smaller network size has also reduced the actual memory consumption.
>
> **Q2**: The proposed is not consistently better than other methods. For those inferior results, some analysis should be provided since the results violate the motivation.
>
> **A** : As we mentioned in the conclusion, our method consistently outperforms other 'fixed-importance' pruning methods. GBN, which outperforms our results, is one of the ‘trainable-importance’ pruning methods. Fixed-importance pruning methods are much more efficient in computational cost and memory usage as trainable-importance pruning methods require training from the whole network, while fixed-importance methods need only to train from a small pruned network, as mentioned in section 4.
>
> For example, we can compare our method with [1], one of the trainable-importance methods, using ResNet architecture on CIFAR-10 dataset. [1] trains the entire network with a sparsity regularizer for 160 epochs, prunes, and finetunes for another 160 epochs. Meanwhile, our method only requires pruning and finetuning on a smaller network for 200 epochs.
>
> For further analysis, we compared our method to [2], which is not a fixed-importance method nor a trainable-importance method, using MobileNetV2 architecture on ImageNet. [2] requires to train a PruningNet, which generates the weights of the pruned network, for 64 epochs. However, the number of parameters of a PruningNet is at least $30$ times larger than the original network. Furthermore, [2]  searches for a well-performing pruned network via an evolutionary procedure, and this search step requires about 1000 evaluations on the test dataset. The whole process before the finetuning step takes about 2 days on a machine with 4 GPUs (RTX-2080 Ti) using PyTorch while our method only takes 2 hours using 10 CPU (Xeon(R) Silver) cores without any GPUs.
>
> Finally, we note that our method has the potential to perform even better by adjusting the normalizing factors. Our experiments on Mobilenet-V2 in supplementary material G show that tuning the normalizing factors can improve the pruning performance. However, we did not conduct an extensive hyperparameter search on this normalizing factor in the main experiments.
>
> **References**
>
> [1] Learning efficient convolutional networks through network slimming, ICCV 17.
>
> [2] MetaPruning: Meta Learning for Automatic Neural Network Channel Pruning, ICCV 19.
>
> [3] NetAdapt: Platform-Aware Neural Network Adaptation for Mobile Applications, ECCV18.

---

### Official Review · AnonReviewer2 · 2020-10-28

**Rating:** 7
**Confidence:** 3

**Review:**

This paper introduces an optimization method for pruning channels in networks. The authors first motivated the proposed approach by showing that current pruning methods will result in "inactive weights" for the following layer.  Then the authors introduce a QCQP optimization method that can constrain the exact amout of resources during the optimization process. Extensive experiments are conducted on different benchmarks with different backbones. And the authors also performed spatial pruning to further reduce resource usage.


####### Strengths######
+ The motivation is clear and the presentation is generally good.
+ The  idea of mitigating the effect of inactive weights is interesting.
+ Extensive studies have been conducted in terms as different datasets/backbones.

####### Weakness######
- The term "the inherent quadratic coupling" used in the abstract is a bit confusing without any explannations.
- I didn't quite follow section 2.2, where the authors discussed the quadratic coupling effect. In figure 2, I understand the prunned channels for the greedy part. But I don't quite get how the proposed approach is able to prune the last channel of the 2nd layer. It would be nice to discuss this when the optimization is introduced?
- Following the previous point, the authors basically are saying QCQP is better than greedy pruning. I couldn't find experimentd comparing these two methods. I mean I understand previous methods are greey based. But it would be nice to have the same implementation by the authors for apple-to-apple comparisons.
- Missing references
[1]Rethinking the Value of Network Pruning
[2] Channel Gating Neural Networks
[3] Dynamic Channel Pruning: Feature Boosting and Suppression


##############Post Rebuttal###############

My concerns are addressed by the authors. I'm keeping my original rating.

---

> ### Author Response · Authors · 2020-11-20
> **Response to Reviewer 2**
>
> We thank reviewer 2 for encouraging comments ("motivation is clear", "idea of mitigating the effect of inactive weights is interesting")  and constructive feedback.
>
> **Q1**: I don't quite get how the proposed approach is able to prune the last channel of the 2nd layer. It would be nice to discuss this when the optimization is introduced?
>
> **A** :  Thank you for pointing this out. In our method, we first find the optimal channel activation ‘r’ and then compute the pruning mask ‘A’ to prune filters. This process directly optimizes our objective and finds the global optimum, avoiding the possible local-optimum solutions from the greedy approach. For example, in figure2, we first find the optimal channel activation $r^{(l-1)}, r^{(l)} ,r^{(l+1)}$ which maximizes the sum of importance of remaining filters. Concretely, $r^{(l-1)}=[1,1,1]^\intercal$, $r^{(l)}=[0,1]^\intercal$, and $r^{(l+1)}=[1,1,0]^\intercal$. $A^{(l)} = r^{(l-1)}{r^{(l)}}^\intercal \otimes J_{K_l}$ lead to
> $A^{(l)} = \begin{bmatrix} 0 & 1 \\\ 0 & 1 \\\ 0 & 1\end{bmatrix} \otimes J_{K_l}$ and $A^{(l+1)} = \begin{bmatrix} 0 & 0 & 0\\\ 1 & 1 & 0\end{bmatrix} \otimes J_{K_{l+1}}$. We clarified these details in section 3.1.
>
> **Q2**: Experiments comparing these two methods (QCQP and greedy approach).
>
> **A** :
> Thank you for the suggestion. For apple-to-apple comparison, we compare QCQP (Ours) to the greedy approach (Greedy) in ResNet-20. Note that Greedy prunes the channels starting from the first layer while removing inactive weights. Also, in Greedy, we prune the channels with a uniform ratio in each layer and adopt a common heuristic to ignore the skip addition. We compared the objective value of Ours and Greedy under several FLOPs constraints (20%, 40%, and 60% of original FLOPs), and present the results below. Ours finds a better optimum compared to Greedy under all FLOPs constraints.
>
> |   FLOPs (%)$\downarrow$   |   Greedy   |     Ours    |
> |:--:|:--:|:--:|
> |20                 |    138.3     |    **196.5**      |
> | 40                |    261.8     |    **361.9**      |
> | 60                |    362.0     |    **473.2**      |
>
> **Q3** : Missing references.
>
> **A** : Thank you for your comment. We added the three references to the related works in the revised version.

---

### Official Review · AnonReviewer5 · 2020-11-06
**Writing is good but with limited novelty.**

**Rating:** 5
**Confidence:** 5

**Review:**

This paper mainly improves the idea of "PRUNING FILTERS FOR EFFICIENT CONVNETS" by encouraging the pruning with a {0-1} optimization instead of a greedy manner. Experiments validate the effectiveness of the proposed method.

Pros:
+ Writing is good, and the technical details seem sound and clear.
+ The motivation makes sense.

Cons:
- The novelty is limited. The formulation of the 0-1 optimization for pruning is simple and intuitive. Concretely, it leverages the pre-trained weights for the unpruned network and tries to select the kernels with the maximum magnitude. For me, I am not sure whether the novelty is up to the standard of ICLR venue.
- The objective is to maximize the norm of selected filters. However, magnitude-based pruning is already challenged for it is not accurate to indicate the selection.
- Authors claim that current pruning papers can not reach a strict constraint for FLOPs during pruning. However, it is not true for recent pruning methods, such as AutoSlim, TAS and MetaPruning. Necessary discussions are needed.
- Pruning on recent compact networks is favoured, such as MobileNetV2, which is also a routine network for many pruning papers.

[ICLR2017] PRUNING FILTERS FOR EFFICIENT CONVNETS
[2019] AutoSlim: Towards One-Shot Architecture Search for Channel Numbers
[ICCV2019] MetaPruning- Meta Learning for Automatic Neural Network Channel Pruning.pdf
[NIPS2019] Network Pruning via Transformable Architecture Search

---

> ### Author Response · Authors · 2020-11-20
> **Response to Reviewer 5 - (1)**
>
> We thank reviewer 5 for the encouraging comments (“Writing is good”, “the technical details seem sound and clear”, “motivation makes sense”) and constructive feedback.
>
> **Q1** : The novelty is limited. The formulation of the 0-1 optimization for pruning is simple and intuitive.
>
> **A** :
> To the best of our knowledge, optimally pruning network channels and shape columns modeling the quadratic coupling between neighboring layers in a principled optimization framework (QCQP) has never been explored before. Our formulation also theoretically certifies that any inactive weights do not exist in our network during the pruning procedure, fundamentally guaranteeing the exact computation of the true objective and target resources such as FLOPs and network size (see Proposition 1 and 2). In contrast, previous fixed-importance pruning methods cannot safely remove inactive weights during the pruning process and resort to post-hoc heuristics to remove those inactive weights **after the pruning procedure**.
>
> Furthermore, our approach can naturally handle nonsequential connections, such as skip additions and skip concatenations, more flexibly. For example, assume the output feature map of a layer (A) and a skip-connected feature map (B) add up to be the input feature map of the next layer (C). Previous pruning method [1] handles the skip connection by simply grouping their channels - each $i$-th channels of A, B, and C are pruned as one. We note that this heuristic is to simplify the pruning procedure, but this severely limits the feasible set of pruned networks that can be discovered. On the other hand, in our optimization framework, pruning the $i$-th channel of one feature map does not necessarily lead to pruning the others’ $i$-th channels. Instead, our optimization framework provides the minimum set of rules a pruning algorithm should follow, which results in more flexibility during the pruning procedure (see supplementary material A.1).
>
> We are unaware of other papers with similar contributions, and it would be helpful if the reviewer could point us to such works.
>
> **Q2**:  Magnitude-based pruning is already challenged for it is not accurate to indicate the selection.
>
> **A** :
> To clarify, we used the term ‘magnitude-based pruning’ to refer to pruning methods based on the importance of neurons or channels. The importance tensor used can be freely replaced with other measures of importance. We believe the proposed method can be used as an analysis tool to compare the performance of several magnitude-based channel pruning methods in the long run.
>
> That said, we note that magnitude-based pruning is an active area of research within efficient network inference. To name a few, [2] provides a method for magnitude-based pruning by considering the effect of neighboring neurons. [3] studies several possible metrics for measuring the importance of neurons. [4] provides a pruning method based on the collaborative importance of neighboring channels in the same layers. [5] iteratively prunes the network according to the average activation of channels.
>
> Furthermore, we explain the efficiency of magnitude-based pruning methods over other pruning methods (AutoSlim, TAS, MetaPruning) in the next question.

---

> > ### Author Response · Authors · 2020-11-20
> > **Response to Reviewer 5 - (2)**
> >
> > **Q3** :  Other recent pruning methods, such as AutoSlim, TAS and MetaPruning can reach a strict constraint for FLOPs.
> >
> > **A** :
> > The reviewer is correct that AutoSlim, TAS, and MetaPruning can also reach the resource constraints strictly. However, these methods are much less efficient compared to our method: 1) AutoSlim needs to train multiple slimming networks. 2) TAS requires extensive evaluations on a large number of different architectures for the neural architecture search. Concretely, it takes 59 hours on 4 GPUs (Tesla V-100) for TAS to prune a ResNet-18 network on ImageNet, while our method can prune a ResNet-50 network on ImageNet in 3 hours only using 10 CPU (Xeon E5-2650) cores without any GPUs (see supplementary material B). 3) MetaPruning trains a PruningNet, which is at least 30 times larger than the original network in terms of network parameters, and then searches a well-performing pruned network via evolutionary methods.
> >
> > Meanwhile, our method is along the line of fixed-importance pruning methods and does not require exhaustive network training like the methods above. Also, our approach can meet the resource constraints tightly with only one round of pruning and finetuning. Other fixed-importance methods need multiple rounds of pruning and finetuning to achieve the same goal.
> >
> > Recently, another paper suggested a budget-aware regularizer to strictly satisfy the target resource constraints ([6]). Our method differs from [6] in that our method deals with target resource constraints in the ‘fixed-importance’ pruning framework, while the work of [6] lies in the ‘trainable-importance’ pruning framework, which is more computationally expensive.
> >
> > **Q4**:  Pruning on recent compact networks is favoured, such as MobileNetV2, which is also a routine network for many pruning papers.
> >
> > **A** : Thank you for the suggestion. We provide the MobileNetV2 experiment results in the table below (please see the revised supplementary material G as well). ‘(tuned)’ indicates that the normalizing factor ($\gamma_{l}$ in the main paper) is tuned with grid search. A fixed value is used otherwise. Our method shows performance competitive to other recent pruning methods, [7], [8], and MetaPruning. However, we note that our method is much more efficient than those methods since [7] requires repetitive finetuning steps on the proposed networks, [8] also requires iterative trial and error steps to train the RL agent, and MetaPruning trains PruningNet, which is at least $30$ times bigger than the original model.
> >
> > |**Network**|**Method**|**Top 1 Pruned Acc$\uparrow$**|**Top1 Acc drop$\uparrow$**|**FLOPs (%)$\downarrow$**|
> > |:---:|:--:|:--:|:--:|:--:|
> > | MobileNetV2  |       [7]               |            70.9           |          0.9          |       70  |
> > |                       |       [8]                |            70.8           |          1.0          |       70   |
> > |                       |   MetaPruning   |            **71.2**           |           **0.6**          |       69 |
> > ||_________________________________|_________________________________|_________________________________|_________________________________|
> > |                       |       ours-c         |            70.8 	 |          1.0          |      **67** |
> > |	             |      ours-cs         |           70.2           |          1.6          |       **67** |
> > |                       |  ours-c (tuned)   |            71.0           |          0.8         |       **67** |
> > |                       | ours-cs (tuned)  |            70.9           |           0.9        |       **67** |
> >
> > **Table** Top1 pruned accuracy and accuracy drop from the baseline network at given FLOPs on MobileNetV2 architecture at ImageNet.
> >
> > **References**
> >
> > [1] Gate decorator: Global filter pruning method for accelerating deep convolutional neural networks, NeurIPS19.
> >
> > [2] Lookahead: A Far-sighted Alternative of magnitude-based pruning, ICLR 2020.
> >
> > [3] Importance Estimation for Neural Network Pruning, CVPR 2019.
> >
> > [4] Collaborative Channel Pruning for Deep Networks, ICML19.
> >
> > [5] DropNet: Reducing Neural Network Complexity via Iterative Pruning, ICML20.
> >
> > [6] ChipNet: Budget-Aware Pruning with heaviside continuous approximations, under review, ICLR 2021
> >
> > [7] NetAdapt: Platform-Aware Neural Network Adaptation for Mobile Applications, ECCV18.
> >
> > [8] AMC: automl for model compression, ECCV18.

---

### Author Response · Authors · 2020-11-20
**Dear Reviewers**

Thank you for your time and the effort spent providing thoughtful feedback. We appreciate the encouraging comments [R1] “Good submission focusing on a valuable topic”, “valuable both in theory and applications”. [R2] “The motivation is clear” and “The idea of mitigating the effect of inactive weights is interesting”. [R3] “well-written and well-motivated”, “simple to use”, and “may inspire following works in this area”. [R5] “Writing is good, and the technical details seem sound and clear”, “The motivation makes sense”.

We address your concerns in the individual replies and update our submission.

---

### Decision · Program_Chairs · 2021-01-07
**Final Decision**

**Decision:**

Reject

**Comment:**

This paper proposed a new optimization framework for pruning CNNs considering coupling between channels in the neighboring layers. Two reviewers suggested acceptance and two did rejection. The main concerns of the negative reviewers are (a) limited novelty, (b) limited performance metrics and (c) limited baselines. The authors' response did not fully clarify the reviewers' concerns during the discussion phase, and AC also agrees that they should be resolved to meet the high standard of ICLR. Hence, AC recommend rejection.

Here is additional thought from AC. The authors propose ours-c and ours-cs. The latter is reported to outperform the former in terms of FLOPs, but AC thinks the former may have merits in other more important performance metrics, e.g., the actual latency and/or memory consumption on a target device. More discussions and results for this would strengthen the paper.